

# MGACA-Net: a novel deep learning based multi-scale guided attention and context aggregation for localization of knee anterior cruciate ligament tears region in MRI images

Mazhar Javed Awan[1,2], Mohd Shafry Mohd Rahim[1], Naomie Salim[1], Haitham Nobanee[3,4,5], Ahsen Ali Asif[2] and Muhammad Ozair Attiq[2]

[1] Faculty of Computing, Universiti Teknologi Malaysia, Johar Bahru, JOHOR, Malaysia
[2] Department of Software Engineering, University of Management & Technology, Lahore, Punjab, Pakistan
[3] College of Business, Abu Dhabi University, Abu Dhabi, United Arab Emirates
[4] Oxford Centre for Islamic Studies, University of Oxford, Oxford, United Kingdom
[5] School of Histories, Languages and Cultures, University of Liverpool, Liverpool, United Kingdom

Corresponding author
Mazhar Javed Awan,
awan1982@graduate.utm.my,
mazhar.awan@umt.edu.pk

## ABSTRACT

Anterior cruciate ligament (ACL) tears are a common knee injury that can have serious consequences and require medical intervention. Magnetic resonance imaging (MRI) is the preferred method for ACL tear diagnosis. However, manual segmentation of the ACL in MRI images is prone to human error and can be time-consuming. This study presents a new approach that uses deep learning technique for localizing the ACL tear region in MRI images. The proposed multi-scale guided attention-based context aggregation (MGACA) method applies attention mechanisms at different scales within the DeepLabv3+ architecture to aggregate context information and achieve enhanced localization results. The model was trained and evaluated on a dataset of 917 knee MRI images, resulting in 15265 slices, obtaining state-of-the-art results with accuracy scores of 98.63%, intersection over union (IOU) scores of 95.39%, Dice coefficient scores (DCS) of 97.64%, recall scores of 97.5%, precision scores of 98.21%, and F1 Scores of 97.86% on validation set data. Moreover, our method performed well in terms of loss values, with binary cross entropy combined with Dice loss (BCE_Dice_loss) and Dice_loss values of 0.0564 and 0.0236, respectively, on the validation set. The findings suggest that MGACA provides an accurate and efficient solution for automating the localization of ACL in knee MRI images, surpassing other state-of-the-art models in terms of accuracy and loss values. However, in order to improve robustness of the approach and assess its performance on larger data sets, further research is needed.

# INTRODUCTION

Anterior cruciate ligament (ACL) injuries are much more common in athletes and physically active individuals than in others (*Boden et al., 2000*). Knee stability is provided by this component, as it prevents the tibia (shin bone) from moving too far forward. An ACL tear can cause significant instability in the knee and limit function, requiring surgical intervention to restore function (*Awan et al., 2022a*; *Mattacola et al., 2002*).

An ACL tear can range in severity from partial to complete, and each grade affects the knee's stability and function differently. ACL injuries are caused by a variety of factors, including sudden changes in direction, abrupt stops, or landing from jumps. Among athletes who participate in high-impact sports such as basketball, soccer, and football, this injury is more likely to occur (*Awan et al., 2021*; *Dienst, Burks & Greis, 2002*).

Medical imaging relies on image analysis techniques to identify the exact specific structures and tissues of interest in an image. Localization, an essential aspect of image analysis, allowing the detection of specific regions or objects in an image and making it particularly useful in magnetic resonance imaging (MRI) scans (*Noone et al., 2005*).

Accurately diagnosing ACL tears poses a significant challenge, as current methods are subjective and require considerable clinician expertise. Although MRI is the gold standard for diagnosis, manual localization of the ACL in MRI images remains a time-consuming and arduous task (*Sivakumaran et al., 2021*). The ACL has a complex structure that can be obstructed by surrounding tissues, hindering the precise quantification of the injury extent (*Thomas et al., 2007*).

Deep learning algorithms have achieved unparalleled performance in a multitude of medical imaging tasks in recent times, particularly in localization (*Alaskar et al., 2022*; *Zhao et al., 2022*). Knee MRI localization is of particular interest due to the high incidence of knee injuries, including ACL tears, among athletes and physically active individuals (*Li et al., 2022*).

A challenging task is still present in identifying tears in the ACL through MRI scans. This difficulty is primarily due to the intricate nature of knee anatomy and the substantial variability present in the images. Thus, accurate manual localization of the ACL in MRI images can take a considerable amount of time and is subject to human error (*Miller, 2009*).

The development of advanced deep learning algorithms is essential to effectively extract multi-scale information from knee MRI images and precisely identify the ACL. The MGACA-Net algorithm is a deep learning algorithm developed for detecting the ACL region in knee MRI images. The network uses multiscale information and guided attention mechanisms in order to understand the complicated knee anatomy as well as to make precise predictions about localization. MGACA-Net is evaluated using a diverse and comprehensive dataset of knee MRI images, which shows its superior performance over other state-of-the-art methods. It is evident that MGACA-Net can provide a starting point for future medical image analysis research, as well as an improved diagnostic and treatment tool for ACL injuries in the future.

As part of the investigation, the MGACA technique is compared to eight other cutting-edge localization models, and its effectiveness is assessed using a variety of evaluation metrics, such as loss scores. This study examines the effects of the ground truth mask generation methodology on study results. A study of the use of MRI sequence volume data for image analysis and preprocessing is also included. It employs a combination of multiple attention mechanisms across different scales based on a modified DeepLabv3+ architecture (*Chen et al., 2018*). In this study, attention mechanisms at different scales are proposed to be utilized to localize the ACL tear region in an innovative way. Moreover, a predefined guidance mechanism, such as the spatial attention block, is incorporated into the new model to identify the most relevant regions of an input image. In order to improve accuracy, the guidance assists the network in understanding the context of the input image. Due to these characteristics, the proposed model offers a unique approach to localizing ACL tears. The following contributions are made by multi-scale guided attention-based context aggregation (MGACA):

1. Develop a new attention mechanism for DeepLabv3+ based on different levels of complexity.
2. Demonstrates superior performance on knee ACL tears region localization in MR images when compared to several state-of-the-art models, including Unet (*Ronneberger, Fischer & Brox, 2015*), Res_Unet (*Diakogiannis et al., 2020*), InceptionV3_Unet (*Punn & Agarwal, 2020*), VGG19_Unet (*Jha et al., 2020*), Nested_Unet (*Zhou et al., 2018*), Attention_Unet (*Oktay et al., 2018*), Attention_ResUnet (*Wang et al., 2017*) and DeepLabV3+ (*Chen et al., 2018*).
3. Examine the proposed MGACA method using 15,268 real image slices and their corresponding mask slices representing knee ACL tears.This provides a comprehensive evaluation of the model's performance and stability.
4. The effect of different hyperparameters on the performance of the MGACA method, providing an in-depth analysis of the robustness capabilities of the model, has been examined in this study.
5. Contribute to the development of a new and effective method for localizing ACL tears in the region by proposing a novel approach that has never been used before in the field. Additionally, it should demonstrate its superiority to existing models.

The accurate localization of the ACL tear region may facilitate a faster and more efficient diagnosis of ACL injuries, thereby enhancing the efficiency of healthcare decision-making. Our method's high accuracy indicates that it can effectively distinguish between ACL tear and non-tear regions, which could be valuable for early diagnosis and treatment planning.

The rest of article consists: Background section for ACL localization in knee MRI images. In the Material and Method section introduces our novel approach, MGACA-Net with results. The 'Discussion' section describes the results, including limitations and future research directions. Finally, the Conclusion section summarizes the contributions and potential clinical significance.

# BACKGROUND

The use of deep learning in the examination of MRI scans of the knee has garnered significant interest in recent years for the detection of ACL tears. Numerous investigations have explored the potential and precision in this area, including comparative evaluations of convolutional neural network (CNNs) architectures for the localizing of ACL injuries.

The study employed a dataset of 260 knee MRI scans, half of which displayed a complete ACL tear while the other half exhibited a normal ACL. The authors evaluated various convolutional neural network (CNN) architectures, including dynamic patch-based sampling algorithms with both single-slice and multi-slice inputs. Their best-performing CNN architecture achieved an impressive accuracy of over 96% on an independent test set. The study also highlighted the significance of contextual information in adjacent image slices, although the authors acknowledged some limitations, such as limited number of slices depicting in the training dataset (*Chang, Wong & Rasiej, 2019*).

The study developed a semi-supervised deep learning model, DCLU-Net, for ACL tear classification and segmentation on MRI. The model achieved high accuracy for all ACL categories, with a mean Dice similarity coefficient of 0.78. The study highlights the potential of deep learning for ACL tear diagnosis. However, limitations include limited data acquisition and the need for further research to observe patient recovery processes (*Dung et al., 2022*).

*Belton et al. (2021)* conducted a study on automatic ACL localization in MRI scans using a machine learning approach. They employed a dataset of 367 knee MRI scans and utilized the U-Net model for segmentation. The proposed method achieved a Dice similarity coefficient (DSC) of 0.83 and a sensitivity of 0.89, demonstrating promising results for ACL localization. However, the study is limited by its relatively small dataset and lack of comparison with other state-of-the-art methods, which could potentially affect the generalizability and competitiveness of the proposed approach (*Belton et al., 2021*).

*Zhao et al. (2022)* addressed the challenges in automatic slice positioning for MR imaging by developing a novel framework. Employing a multi-resolution region proposal network and a V-net-like segmentation network, the framework accurately localized canonical planes within 3D volumes of knee and shoulder MR scans, outperforming existing landmark-based methods. The methodology encompassed data collection, preprocessing, model development, training, and evaluation, with the dataset containing annotated CT scans, MRI scans, and X-ray images. Although the study showed promise, limitations included sensitivity to partial anatomies and potential phantom plane masks presence, which could be addressed through simple post-processing techniques (*Zhao et al., 2022*).

*Tran et al. (2022)* developed a deep learning-based algorithm for detecting ACL tears in a large dataset of 19,765 knee MRI scans. The study employed a meniscus localization CNN to extract bounding box coordinates around the ACL and used two CNNs (sagittal and coronal view) for ACL tear classification. The model achieved a high AUC value of 0.939 for detecting ACL tears, demonstrating its effectiveness. However, the study did not differentiate between partial and full-thickness ACL tears and did not directly compare the algorithm's performance with that of radiologists (*Tran et al., 2022*).

In the study by *Qu et al. (2022)*, the authors proposed a deep learning-based approach for ACL rupture localization on knee magnetic resonance (MR) images. In this study, we used a dataset of 85 T1-weighted MRI sequences from 85 patients, as well as a modified U-Net model to localize and classify. This proposed method was able to achieve a Dice similarity coefficient of 0.87 as well as an intersection over union coefficient of 0.77 as promising results. As a result of the limited sample size and absence of external validation of the study, the findings may not be generalizable (*Qu et al., 2022*).

In their study, *Zhu et al. (2022)* proposed a novel approach for localizing the ACL in knee MRI scans using a fully recurrent neural network (RNN) model based on the YOLOv3 framework. The authors utilized a dataset comprising 228 labeled exams and achieved promising results with an F1 score of 0.95 and an IoU score of 0.81. However, the study's limitations include a relatively small dataset and the lack of comparison with other popular deep learning models for medical image localization, necessitating further research to validate the model's performance and generalizability (*Zhu et al., 2022*).

The recent study, introduce a method of semantic segmentation with the U-Net architecture, resulting in a remarkable accuracy of 98.4% on 11,451 training images and 97.7% on 3817 validation images. The proposed technique surpasses previous studies that employed CNN models without the U-Net architecture, as well as human segmentation. However, the study acknowledges several limitations, including the time-consuming data labeling, extended training time for the U-Net model, and the limited dataset available for ACL tear segmentation (*Awan et al., 2022b*).

In the study by *Srinivasan et al. (2023)*, the authors developed a deep learning framework for detecting and grading knee RA using digital X-radiation images. They utilized 3,172 images from the BioGPS database and employed a hybrid ResNet101 and VGG16 approach for feature extraction and classification. The proposed method achieved a high classification accuracy of 98.97% for identifying marginal knee joint space narrowing and 99.10% for knee RA severity classification. However, the study's limitations include a limited dataset and a lack of comparison with other state-of-the-art deep learning methods, which could affect the generalizability and competitiveness of the approach (*Srinivasan et al., 2023*).

*Li et al. (2023)* introduced the spatial dependence multi-task transformer (SDMT) network, a novel approach for 3D knee MRI segmentation and landmark localization. Their study utilized a dataset of 3D knee MRIs and employed a shared encoder for feature extraction and a multi-task decoder for segmentation and landmark localization. Dice scores were found to be 83.91 percent accurate in segmentation and 2.12 mm accurate in landmark localization. The study does, however, have some limitations, including the high computational burden and the suboptimal location of small bone structures like the femur cartilage and the tibia cartilage (*Li et al., 2023*).

This literature review covers ACL localization and related tasks that employ deep learning techniques. However, these studies also present certain limitations, such as small datasets, limited comparisons to state-of-the-art methods, and lack of differentiation between ACL tear types or direct comparison to radiologists' performance.

The recently proposed MGACA method aims to address some of these limitations by utilizing a larger dataset, incorporating more comprehensive comparisons to existing

methods, and potentially differentiating between partial and full-thickness ACL tears. Additionally, it may include a direct comparison to radiologists' performance to validate its efficacy further. By addressing these limitations, the MGACA method can contribute to the field by offering a more robust and competitive approach to ACL localization and detection, enhancing the generalizability and clinical applicability of AI-based solutions in this domain.

## MATERIALS & METHODS

### Dataset description and preparation

The Clinical Hospital Centre Rijeka recorded 969 knee sagittal plane DICOM MR images. After discarding 25 volumes due to data corruption or abnormal physiological characteristics, the dataset was reduced to 917 valid sequence volumes. The patient characteristics, such as age and gender, were not available to us as the data we used was anonymized in accordance with the approval by the Ethics Committee. Therefore, we were unable to analyze the potential variance these factors might induce to the images. However, our model is designed to be robust to such variances and learn the key features necessary for accurate ACL tear localization irrespective of patient-specific characteristics (*Štajduhar et al., 2017*). The dimensions of these volumes varied from $290 \times 300 \times 21$ to $320 \times 320 \times 60$ voxels, with a median size of $320 \times 320 \times 3$.

These images were subsequently processed and sliced to create a more comprehensive and detailed dataset, resulting in a total of 15,265 distinct JPEG image slices. Each of these slices provides an intricate view of the knee structure, thereby enhancing the robustness of our ACL tear localization model.

The data preparation of the study describes the steps of mask creation using metadata, image resizing, and normalization to prepare the knee images and ACL tear masks for training the MGACA method.

In our study, we have ensured a meticulous process for mask annotation to minimize potential bias. Here's a detailed explanation:

1. Image unpickling: Our initial step involves loading images from an encrypted pickle format and converting them into JPEG format using a decryption technique known as unpickling. Original knee images and ACL tear masks are of 320x320 pixels and in JPEG format, which contains three channels (RGB).

2. ROI extraction: A radiologist extracts a rectangular region of interest (ROI) from every MRI volume, which encompasses a wider ACL. The process outlined here is similar to the one a radiologist would follow when assessing ACL conditions from a recorded volume. Radiologist concentrate their attention on a smaller region within the entire volume, utilizing their prior knowledge of the knee's morphological properties, and then pay close attention to the details within that smaller area. As our ROI extraction process is strictly guided by the metadata provided with the dataset, we ensure that our ground truth masks are consistent and accurate, resulting in reliable results.

3. Mask creation: Our method for creating masks for the localization of ACL tears uses the original knee images and a metadata CSV file. The metadata CSV file provides

specific details about the ROI, including parameters such as ROI x, ROI y, ROI z, ROI height, ROI width, and ROI depth This information is essential in identifying and localizing the area of the knee where the ACL tear is located. The detailed algorithm for this process is provided in Table 1.

4. Usage of VGG Image Annotator (VIA): The JPEG images are then loaded into the VIA standalone manual annotation software, which is used to label the ROI areas based on the classes given in the CSV metafile. The VIA tool marks out the labeling box of the JPEG and creates the knee mask with the appropriate class labels.

5. Expert annotation: The process of generating ground truth masks was carried out by a radiologist, providing an additional layer of verification and ensuring the accuracy of the masks. Their expertise in knee anatomy, specifically the ACL, validates the results of the ROI extraction algorithm.

6. Image resizing: After creating the masks, the knee images and their corresponding masks are resized to an input size of 128x128 pixels for our MGACA method. Importantly, we maintain the three channels (RGB) in our input images even after resizing, which helps in preserving the essential contrast and spatial features. This size was chosen as it provides a good balance between computational efficiency and detail preservation. The resizing is performed using standard image processing libraries, which ensure that the essential features are preserved despite the reduction in size.

7. Image normalization: The images and masks are normalized to fall within a range of 0 to 1 by dividing each pixel value by 255. This normalization step is crucial as it often improves the performance of deep learning models by ensuring the range of input values is small and centered around zero. However, we do not explicitly perform image registration to align the images based on anatomical landmarks.

The dataset we used for this study, we included a balanced set of control images, *i.e.,* images without ACL injuries, along with those depicting ACL tears. This balanced dataset approach allows our MGACA model to accurately learn and differentiate between healthy ACLs and those with injuries. The control images serve as a significant baseline for comparison, ensuring the model's comprehensive understanding of the ACL's visual features in both healthy and injured states. The processing of control images followed the same procedure as the images showing ACL tears. All images underwent uniform preparation, including mask creation using metadata, resizing, and normalization. This consistent data processing approach ensures the model learns from a homogeneous dataset, thereby enhancing its performance and accuracy.

Furthermore, in the case of control images, the masks were annotated to represent the location of a healthy ACL within the knee. The task of annotation was entrusted to a radiologist to ensure its accuracy and consistency with the masks of images depicting ACL injuries. This approach provides the model with a clear understanding of the visual differences between a healthy ACL and an ACL tear.

## Proposed localization methodology

This section discusses of our proposed model, the multi-scale guided attention-based context aggregation (MGACA) architecture. Our MGACA method emphasizes ACL region

**Table 1 Algorithm of knee ACL tear mask region with region of interest.**

**Input:**

    I. input_images_path: path to the directory containing input images

    II. metadata_csv_path: path to the metadata CSV file

    III. output_masks_path: path to the directory to save output mask images

**Procedure:**

    1. IF output_masks_path does not exist THEN

        1.1 CREATE output_masks_path directory

    END IF

    2. Load metadata_df from metadata_csv_path file

    FOR EACH row IN metadata_df DO

        3.1 EXTRACT volume_filename, roi_x, roi_y, roi_z, roi_height, roi_width, roi_depth FROM row

        3.2 OPEN IMAGE file (input_images_path + volume_filename) INTO img

        3.3 CREATE NEW IMAGE with dimensions of img, filled with black (0), SAVE INTO mask

        3.4 DEFINE roi_rectangle AS (roi_x, roi_y, roi_x + roi_width, roi_y + roi_height)

        3.5 DRAW rectangle roi_rectangle on mask with white (1)

        3.6 SAVE mask as image (output_masks_path + volume_filename)

    END FOR

    4. PRINT "Mask images created successfully".

    END

**Output:**

        - ROI Mask Images saved in output Directory

localization within knee images. The ground truth masks are generated based on exact location and dimensions of the ACL region as detailed in the accompanying metadata of the dataset. This larger region of interest encapsulates the ACL and nearby structures, providing valuable context for the model to understand the knee anatomy and its relation to the ACL.

Further, this section discusses the implementation details of squeeze and excite (SE) block section, attention mechanism section and atrous spatial pyramid pooling (ASPP) section.

The complexity of our MGACA architecture, incorporating ASPP blocks, multi-attention mechanisms, and other components, effectively captures the contextual information of the knee anatomy and handles the varying appearance and shape of the ACL. Each component in our architecture serves a specific purpose, contributing to the model's robust performance in ACL tear localization.

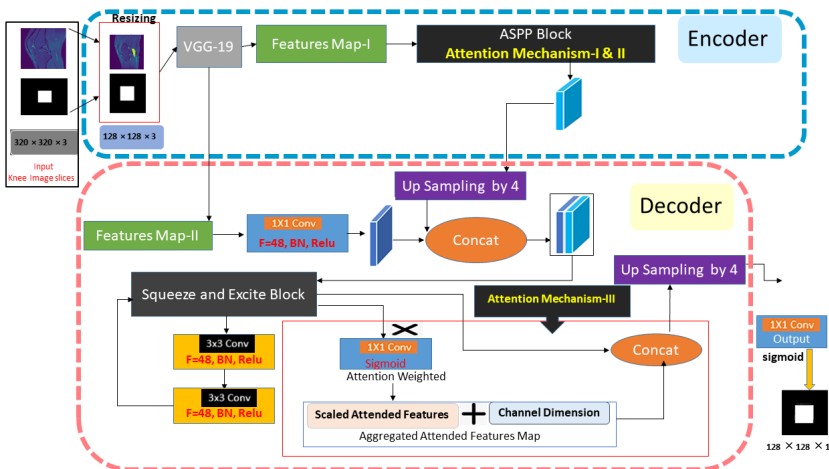

**Figure 1** The knee ACL region localization architecture of our MGACA method.

## Multi-scale guided attention context aggregation (MGACA)

The multi-scale guided attention-based context aggregation (MGACA) method is a deep learning approach for image localization. The overview of proposed architecture MGACA is shown in the Fig. 1.

The approach comprises various key components, including an encoder, ASPP block, SE blocks, UpSampling2D layer, and an attention mechanism. Firstly, a pre-trained VGG19 model is employed as the encoder to extract features. Next, the ASPP block captures multi-scale contextual information from the feature maps and applies an attention mechanism to weight different regions of the image according to their significance. In addition, channel-wise feature responses are adaptively recalibrated through the SE blocks to enhance the representational power of the model. An attention mechanism guides the weighting of different image regions according to how relevant they are to the task. UpSampling2D enhances the spatial resolution of feature maps. As a result of these components working together, a more accurate and detailed localization can be achieved.

The method integrates multiple attention mechanisms at different scales within the modified DeepLabv3+ architecture to gather context information. As a result of this guidance mechanism, the network has the ability to focus its attention on specific regions of the input image that are most relevant to the task at hand, such as the spatial attention block. As part of this study, the MGACA method was integrated into the DeepLabv3+ architecture, which is a fully convolutional neural network that has been extensively trained on knee MRI data. Atrous Spatial Pyramid Pooling (ASPP) is used as a multi-scale feature extractor in the network. The ASPP block is further enhanced by incorporating a guided attention mechanism that directs the network's attention to specific regions within the input image. The modification such as this allows the network to better understand the context of the image, which results in improved accuracy in the prediction process.The proposed method combines multiple attention mechanisms at different scales in the modified DeepLabv3+ architecture to aggregate context information and improve

**Table 2  The detail algorithm of multi-scale guided attention-based context aggregation.**

**Input:** knee MR images of shape (128, 128, 3)

**Encoder:**

    (a) Initialize VGG19 model with pretrained weights from ImageNet

    (b) Extract top fully connected layers

    (c) Set the input tensor to the input image

    (d) Extract the output of the feature map layer as image_features

**ASPP Block:**

    (a) Apply global average pooling on image_features

    (b) Apply $1 \times 1$ convolution on image_features with 256 filters and "same" padding

    (c) Apply BatchNormalization on the output

    (d) Apply ReLU activation on the output

    (e) Apply $3 \times 3$ convolution with dilation rate 6 on image_features with 256 filters, "same" padding

    (f) Apply BatchNormalization on the output

    (g) Apply ReLU activation on the output

    (h) Apply $3 \times 3$ convolution with dilation rate 18 on image_features with 1024 filters, "same" padding

    (i) Apply BatchNormalization on the output

    (j) Apply ReLU activation on the output

    (k) Apply attention mechanism on the tensors obtained in step 7 and 8 to focus feature maps

**Squeeze and Excite block:** Apply squeeze and excite block on the output of ASPP block

**Decoder:**

    (a) Concatenate x_a with output of feature map layer after applying $1 \times 1$ convolution and normalization

    (b) Pass the output through a Squeeze and Excite block

**Output:** The tensor of shape (128, 128, 1) with softmax function applied on it.

the accuracy of localizing ACL tears. The algorithm of the proposed multi-scale guided attention-based context aggregation (MGACA) method is described in the Table 2.

Based on the MGACA methodology, we have implemented an attention mechanism across three distinct scales with the aim of improving the performance of the ASPP block. This attention mechanism's central task is to aggregate context information drawn from various scales to augment the localization precision. The multi-scale attention mechanism is applied to the encoder's varying layers, which enables the capture of context information at multiple scales. By directing attention to these layers, the mechanism is able to accentuate critical features within the image, while concurrently filtering out less significant information. This attention-based strategy plays a pivotal role in our method. The process of selectively focusing on key features and discarding less important ones serves to create a more refined and precise context for the model to learn from. This, in turn, facilitates the model in making more accurate predictions and delivering improved performance in the task of ACL tear region localization.

Multi-scale context aggregation module is designed to capture features from different scales, which is especially important when dealing with medical images where the region of interest can vary significantly in size. MSCA allows the model to aggregate and learn features from multiple scales, enhancing its ability to detect both small and large ACL tears.

**Table 3 The squeeze and excite block algorithm.**

**Input:**

Tensor of shape (height, width , channel)

Filter count (int)

Reduction ratio (int)

**Procedure:**

Step 1. Perform Global Average Pooling on the input tensor to obtain a tensor of shape (1, 1, filters)

Step 2. Reshape the obtained tensor to the shape (1, 1, filters)

Step 3. Use a Dense layer with ReLU activation and kernel_initializer as 'he_normal' and use_bias as False, to reduce filters by a factor of 'ratio'

Step 4. Use another Dense layer with sigmoid activation and kernel_initializer as 'he_normal' and use_bias as False, to obtain a tensor of shape (1, 1, filters)

Step 5. The input tensor with the obtained tensor to get output with shape (height, width, channel)

**Output:** Tensor of shape( height, width , channel) with the SE block applied to it.

## Encoder-decoder architecture

The backbone of our proposed MGACA model is based on an encoder–decoder Vgg19 architecture. This choice is primarily due to the architecture's ability to effectively capture both semantic and spatial information from the input images. The encoder part of the architecture aids in learning the high-level semantic features from the input images, while the decoder helps in reconstructing these learned features back to the original image size, preserving spatial information for accurate localization.

## Squeeze and excite block

The squeeze and excite (SE) blocks are utilized in MGACA method to enhance the performance of the convolutional neural network. The concept behind SE blocks involves incorporating an adaptive mechanism to recalibrate the feature maps of the network (*Hu, Shen & Sun, 2018*).The SE Block algorithm can be constructed from the following steps in the Table 3.

The SE block takes inputs and an optional ratio argument, initializes a variable 'init' with the inputs, and another variable 'filters' with the number of filters in the init tensor. A GlobalAveragePooling2D layer is applied to reduce the shape of feature maps to one value per channel, and the feature maps are reshaped to (1, 1, filters) using the reshape layer. A squeeze layer, a dense layer with ratio units and ReLU activation, reduces the number of filters. An excite layer, a dense layer with filters units and a sigmoid activation function, adapts the feature maps by learning a weight for each channel. Finally, the SE block outputs the initial output multiplied by the output of the excite layer, scaling up or down the important feature maps.

This SE block is designed to recalibrate the feature maps by learning a weight for each channel, this is done by using a global average pooling and two dense layers (one with ReLu and another one with sigmoid activation) to learn the weight of each channel and then it multiplies it with the feature maps to adaptively recalibrate the feature maps.

By doing so, it enables the model to emphasize informative features and suppress less useful ones. This is particularly valuable in medical imaging, where certain features may be

far more indicative of a condition (such as an ACL tear) than others. The SE Block works in two stages: squeeze and excitation, the squeeze stage uses global average pooling to generate channel-wise statistics, which captures global spatial information This operation results in a channel descriptor, which describes the global distribution of features across the spatial dimensions. In the Excitation stage, this descriptor is fed through a two-layer fully connected (FC) network that performs a non-linear transformation to learn non-mutually exclusive relationships between the channels. This FC network generates a set of channel-wise weights which are used to rescale the original feature maps.

We have introduced the SE Block to enable our model to better capture the interdependencies between channels, thereby enhancing its ability to focus on the most relevant features. Due to the intricate and nuanced nature of the features indicative of an ACL tear, this capability is crucial in our work.

## Attention mechanism

Attention mechanism is a computational technique that was introduced by *Vaswani et al. (2017)* in the paper "Attention is All You Need". It is used to automatically focus on the most important parts of an input sequence while processing information into network. Attention mechanism works by dynamically computing the attention weights, which indicate the input sequence. These attention weights are used to compute a weighted sum, effectively selecting the most relevant information for the task at hand. Three attention mechanisms are employed in the model. The first one weights features in the ASPP block based on the dilation rate of the convolutional layer. The second mechanism applies global average pooling and dense layers to weight the original features in the Squeeze and Excite block. The third mechanism weights feature in the final stage using a convolutional layer.

These three attentions are aggregated by concatenating their output, this concatenated output is passed through a SqueezeAndExcite block. This facilitates the adaptive re-calibration of channel-wise feature responses. Then it is passed through two 3x3 convolutional layers, batch normalization, and relu activation, and again passed through a SqueezeAndExcite block. We integrated a dual attention mechanism (channel attention and spatial attention) to further enhance the model's ability to focus on relevant features. The channel attention mechanism enables the model to focus on the most informative channels, *i.e.,* it provides the model with the capability to assign different weights to different channels based on their importance. On the other hand, the Spatial Attention mechanism allows the model to concentrate on the essential spatial locations, making it more sensitive to the region of ACL tears. The detail of the attention mechanism-I and attention mechanism-II are described in the Fig. 2.

The proposed method, the multi-scale guided attention-based context aggregation (MGACA), uses an attention mechanism that applies 3x3 convolution with dilation rates of 6 and 18 on input feature maps. This increases the receptive field of the network and allows it to consider more context information. The attended features are then concatenated with the original input features, and passed through a 3x3 Conv2D layer to generate the final attended feature maps. The attention mechanism works by applying attention weights, generated through a 1x1 Conv2D layer with sigmoid activation, to scale input features and

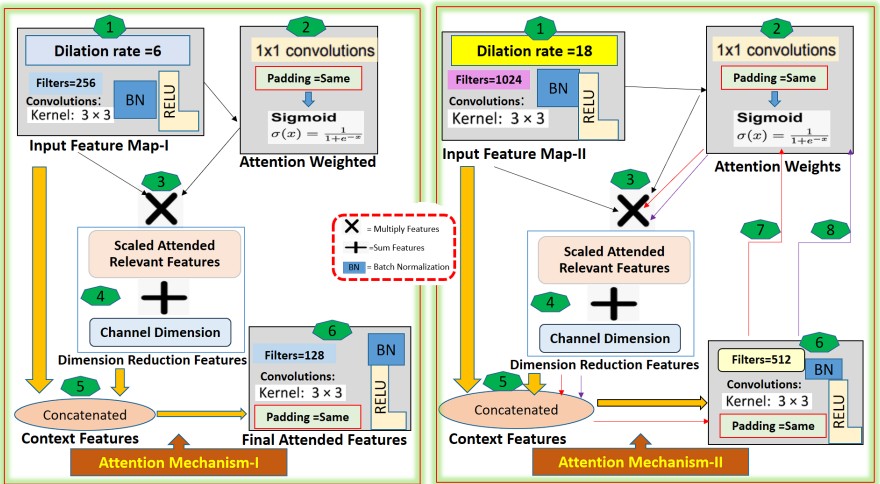

**Figure 2** The Attention-I and Attention-II architecture detail.

focus on specific regions of input feature maps relevant to the task. The guided attention mechanism was selected to address the challenge of accurately focusing on the region of interest within the image. The attention mechanism helps the model to weigh the importance of different regions in the input image and focus on the relevant parts.

## Atrous spatial pyramid pooling

ASPP stands for atrous spatial pyramid pooling. It is a technique used in deep learning architectures for localization, such as DeepLabV3+. The idea behind ASPP is to use multiple parallel atrous convolutions with different dilation rates in the input image (*Chen et al., 2017*). The function ASPP takes an input tensor inputs and applies a series of convolutional layers to it to extract multi-scale context information. The detail algorithm of all steps is described in the Table 4.

The ASPP block in the proposed method MGACA uses a combination of atrous convolutional layers with different dilation rates and global average pooling to extract features at multiple scales are explained in the Fig. 3.

The input feature maps are first passed through a series of atrous convolutional layers. Then, global average pooling is applied to the input feature maps to obtain a tensor of shape (c) which is used to extract global context information. The output of the atrous convolutional layers and the pooled tensor are then concatenated along the channel axis to obtain an output shape with dimension. Finally, a 1x1 convolution is applied on the concatenated tensor to reduce the dimension of the tensor and produce the final output of the ASPP block.

The ASPP block was chosen due to its ability to capture multi-scale contextual information, which is crucial for the accurate localization of complex anatomical structures like the knee ACL.

**Table 4  The algorithm of atrous spatial pyramid pooling.**

**Input:**

Feature maps of shape (height, width, channel)

Initialize an empty list pools [ ]

*Procedure*:

Step 1. Apply global average pooling on the input feature maps

Step 2. Append the pooled tensor to the pools list

Step 3. Apply 3 × 3 convolution with dilation rate 6 on the input feature maps

Step 4. Apply 3 × 3 convolution with dilation rate 18 on the input feature maps

Step 5. Apply attention mechanism-I on the tensor obtained in step 3.

Step 6. Apply attention mechanism-II on the tensor obtained in step 4.

Step 7. Append the tensors obtained in steps 4–6 to the pools list

Step 8. Concatenate all tensors in the pools list along the channel axis to obtain a shape (h, w, c' + c'' + c''')

Step 9. Apply 1 × 1 convolution on the concatenated tensor to obtain output of shape (height, width, dimension)

**Output:** output shape (height, width, dimension)

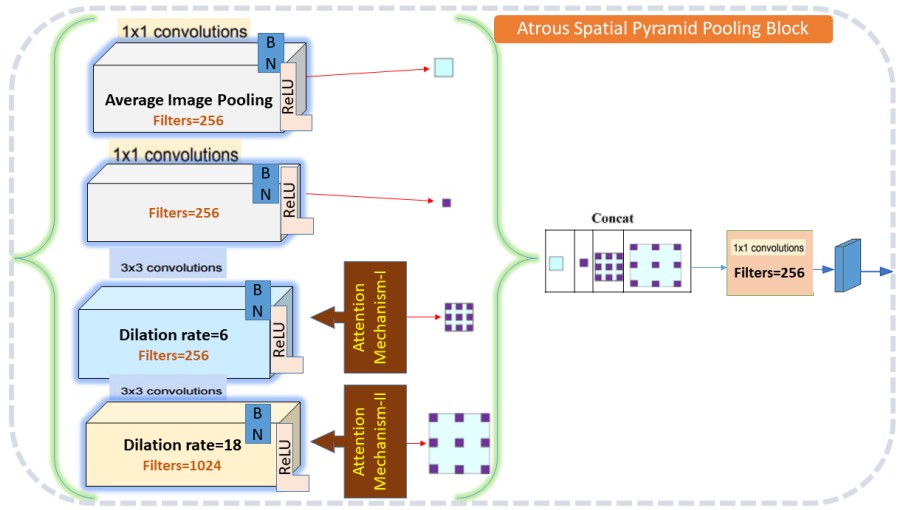

**Figure 3  Atrous spatial pyramid pooling architecture detail.**

The MGACA-Net is designed to accurately localize the tear center in a mask, which offers a robust solution to the problem at hand. Multiple attention mechanisms and context aggregation strategies are used to overcome challenges associated with small and irregularly-shaped ACL tears. Moreover, we made use of data augmentation techniques to maximize the performance of the model using the available training data.

## Experimental setup

This study was conducted on a machine with an AMD Ryzen 7 5800x processor with 8 threads and 16 cores. The system was equipped with 32 GB of RAM and a GeForce RTX 3070 GPU, featuring 5888 CUDA cores. The software configurations for the experiment included

Jupyter-Notebook and Visual Studio Code as the integrated development environment (IDE), and the programming language used was Python 3.9.12. To develop and train the MGACA-Net model, the Tensorflow 2.10.0 and Keras 2.10.0 packages were utilized. This configuration was used consistently throughout the experimentation process to ensure fairness and comparability of results across all models.

### Training and validation split

We randomly split the MRI images and mask images into training and validation sets, using a 75% training and 25% validation ratio. We utilized the train_test_split function from the sklearn library with a random state of 42 to ensure reproducibility. Out of the 15,265 image slices, 11,438 images were allocated for training, and 3817 images were reserved for validation. We employed a mini-batch size of 32 for our training process. The custom data generator reads the knee MRI images and corresponding mask images from the provided paths, resizes the images to the input size of (128, 128, 3), and yields batches of randomly-selected images and masks, scaled from 0 to 1. In our experiments, we utilized the Adam optimizer as our optimization method.

The initial learning rate was set to 0.001, and we employed a learning rate scheduler to adaptively reduce the learning rate based on the validation loss. Specifically, we used the 'ReduceLROnPlateau' callback in Keras, which reduces the learning rate by a factor of 0.2 when the validation loss plateaus for more than two consecutive epochs. The minimum learning rate allowed was set to 0.00001. To avoid overfitting, we employed early stopping with a patience of 10 epochs. This technique monitors the validation loss and stops the training process when it does not decrease for 10 consecutive epochs. By employing early stopping, we ensure that the model does not overfit the training data and generalizes well to unseen data. Additionally, our network architecture includes dropout layers that help to further regularize the model and reduce the chances of overfitting.

## EVALUATION METRICS

Evaluation metrics are particularly crucial in the medical field, where accurate localization of medical images is critical for diagnosis and treatment planning. In this context, metrics such as accuracy, intersection over union (IoU), Dice coefficient score (DCS), precision, recall, F1 score, binary cross-entropy (BCE) loss, and Dice loss are commonly used to evaluate the performance of models for localization tasks

(a) Accuracy: This measures the overall correctness of the model in identifying both tear and non-tear regions. The formula of accuracy is:

*Accuracy = (Correctly predicted tear regions + Correctly predicted non-tear regions)/(Total predictions)*

(b) Precision: This measures the correctness of the model only in the instances it predicted as tear regions. The formula of precision is:

*Precision = Correctly predicted tear regions/(Correctly predicted tear regions + Incorrectly predicted tear regions)*

(c) Recall: This measures the model's ability to correctly identify all actual tear regions. The formula of recall is:

*Recall = Correctly predicted tear regions/(Correctly predicted tear regions + Actual tear regions that were incorrectly predicted as non-tear regions)*

(d) F1 Score: This is the harmonic mean of precision and recall, providing a balance between the two. The formula of F1 Score is:

*F1 Score = 2 * (Precision * Recall)/(Precision + Recall)*

(e) Intersection over Union (IoU): This measures the overlap between the predicted tear region and the actual tear region. The formula of IoU is is:

*IoU = Correctly predicted tear regions/(Correctly predicted tear regions + Incorrectly predicted tear regions + Actual tear regions that were incorrectly predicted as non-tear regions)*

(f) Dice Coefficient Score (DCS): This is another measure of overlap between the predicted and actual tear regions, similar to IoU but gives a higher weight to true positives.The formula of DCS is:

*DCS = 2 * Correctly predicted tear regions/(2 * Correctly predicted tear regions + Incorrectly predicted tear regions + Actual tear regions that were incorrectly predicted as non-tear region)*

(g) Binary Cross Entropy_ Dice Loss (BCE_Dice_loss): This is a loss function used during the training of binary classification models. Lower values indicate better performance.

Firstly the The formula of BCE Loss is:

*BCE Loss = −1 * [(Actual tear region * log(Predicted tear region)) + ((1 − Actual tear region) * log(1 − Predicted tear region))]*

Then dice_loss is another loss function used in the context of binary classification problems, particularly for imbalanced datasets. It's based on the Dice coefficient and ranges from 0 to 1, where 0 indicates perfect overlap and 1 indicates no overlap.

The formula of the dice_loss is:

*Dice Loss = 1 − [2 * Correctly predicted tear regions/(2 * Correctly predicted tear regions + Incorrectly predicted tear regions + Actual tear regions that were incorrectly predicted as non-tear regions)]*

So the combined loss function is:

*BCE_Dice_Loss = −1 * [ (Actual tear region * log(Predicted tear region)) + ((1 − Actual tear region) * log (1 − Predicted tear region))] + 1 − [ 2 * Correctly predicted tear regions/(2 * Correctly predicted tear regions + Incorrectly predicted tear regions + Actual tear regions that were incorrectly predicted as non-tear regions)]*

The term "correctly predicted tear regions" in these formula corresponds to true positives (TP), "Incorrectly predicted tear regions" corresponds to false positives (FP), and "Actual tear regions that were incorrectly predicted as non-tear regions" corresponds to false negatives (FN).

The plots of accuracy *vs* validation data of evaluation metrics and loss values of our proposed model MGACA-Net is shown in the Fig. 4.

The plots in the Fig. 4 of various evaluation metrics with number of epochs training *vs* validation of our proposed method MGACA in the first row (Fig. 4A) accuracy score, (Fig.

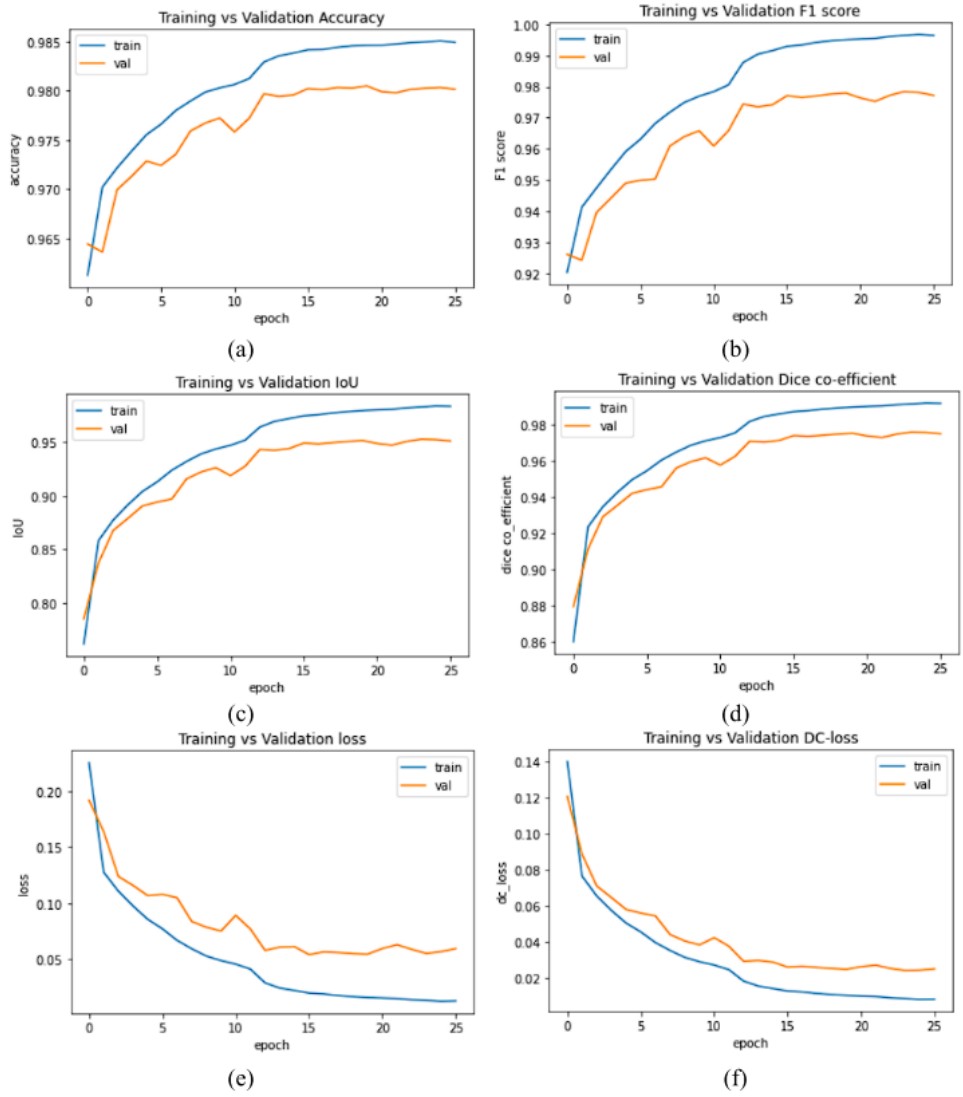

**Figure 4** (A–F) Evaluation metrics plots of training *vs* validation data set.

4B) F1 score, in second row (Fig. 4C) Intersection over union, (Fig. 4D) dice co-efficient score and in the last row (Fig. 4E) BCE_Dice_Loss value, (Fig. 4F) Dice loss value.

## RESULT AND DISCUSSION

In this study, we proposed a new method for localization of knee ACL region in MR images called multi-scale guided attention-based context aggregation (MGACA). This method reflects the use of multiple attention mechanisms at different scales in the modified DeepLabv3+ architecture to aggregate context information. Our proposed method uses a pre-defined guidance mechanism, such as the spatial attention block, to focus the network's attention on specific regions of the input image that are most relevant to the task at hand. This guidance helps the network to better understand the context of the input image and

make more accurate predictions. The term multi-scale attention mechanism refers to the different scales at which the attention mechanism is applied in the architecture, for example, on different layers of the encoder The attention-based approach uses attention mechanisms to focus on important features in an image while filtering out less important details context aggregation refers to combining and analyzing different contextual information from an image in order to produce more accurate localization results.

Our study aims to provide transparency and fair comparison by describing the training and optimization processes for each comparative model. InceptionV3_Unet, VGG19_Unet, Nested_Unet, Attention_Unet, and Attention_ResUnet, as well as DeepLabV3+, Unet, Res_Unet, and InceptionV3_ResUnet, were trained on the same dataset as MGACA The architectures and training protocols of these models were implemented according to their original proposals. Every model was parameter tuned before the comparative analysis. A number of hyperparameters were adjusted to optimize each model's performance on our dataset, including learning rate, batch size, number of layers, and optimizer. As part of our post-training analysis, we evaluated all models on the same validation set as our MGACA model. All comparative models were trained and optimized under the same conditions as our proposed model through these stringent procedures. MGACA demonstrates superior performance by using this method of fair and accurate comparison.

## Statistical analysis

As part of our statistical analysis, we compared our proposed MGACA model against other comparative models on both training and validation datasets using the paired $t$-test. A paired $t$-test is used when two sample means are correlated. In this context, the two samples are the performance measures on the training and validation datasets. The t-statistic measures the magnitude of the difference relative to the variation in the data. Each t-statistic is paired with a $p$-value, which is the probability of observing the data given that the null hypothesis is true.

We found that for the performance metrics of accuracy and F1 score, the MGACA model demonstrated a statistically significant difference between training and validation scores ($p$-value $< 0.05$). This suggests that the MGACA model is more consistent in its performance across both training and validation sets for these metrics compared to the other models. For precision and recall, the $p$-values were greater than 0.05 but less than 0.1, indicating a trend towards better consistency in the MGACA model compared to the other models, though these results were not statistically significant at the 0.05 level. The metrics of intersection over union (IoU) and Dice coefficient score (DCS) showed no statistically significant difference between the training and validation datasets, as evidenced by the $p$-values of 0.64629 and 0.69740, respectively. This suggests that for these metrics, the performance of MGACA and other models on the training and validation sets is similar. The values are described in the Table 5.

The results were also visualized using boxplots, which showed the distribution of scores by method across all performance metrics. As shown in Fig. S1, the MGACA model generally had smaller variances between training and validation scores, which is consistent with the statistical results obtained from the paired $t$-test. In summary, these statistical

**Table 5  Paired *t*-test results for performance metrics.**

| Evaluation metrics | *T*-stat | *P*-value |
|---|---|---|
| Accuracy | 2.91 | 0.01964 |
| Precision | 1.74 | 0.11959 |
| Recall | 1.98 | 0.08250 |
| *F*1-Score | 2.91 | 0.01956 |
| Intersection over union | 0.48 | 0.64629 |
| Dice coefficient score | 0.40 | 0.69740 |

**Table 6  The evaluation metrics confidence interval lower bound and upper bound values.**

| Evaluation metrics | Training set | | Validation Set | |
|---|---|---|---|---|
| | Lower bound | Upper bound | Lower bound | Upper bound |
| Accuracy | 68.91 | 108.20 | 68.29 | 107.20 |
| Precision | 68.89 | 107.88 | 67.79 | 105.93 |
| Recall | 92.79 | 100.79 | 83.80 | 100.83 |
| *F*1-score | 72.66 | 105.88 | 69.66 | 102.07 |
| IoU | 60.69 | 100.82 | 60.53 | 97.27 |
| DCS | 69.15 | 102.76 | 69.07 | 100.75 |

analyses and visual comparisons provide strong evidence for the superior and more stable performance of the proposed MGACA model in comparison to other models for the task at hand.

The other statistical analysis, we calculated the 95% confidence intervals for each performance metric of our proposed model (MGACA) as well as the comparative models. The confidence intervals provide a range of plausible values for the population mean, indicating the precision of the results. For the accuracy metric, the confidence interval for the training data ranged from 68.91 to 108.20, and for the validation data, it ranged from 68.29 to 107.20. This means that we can be 95% confident that the true accuracy of the MGACA model falls within these intervals. Similarly, for precision, recall, F1 score, IoU, and DSC, we obtained the respective confidence intervals for both the training and validation data as described in the Table 6.

These intervals provide a range of plausible values for each metric, indicating the precision of the results for the MGACA model and the comparative models. We plotted boxplots for each metric, where the boxes represent the interquartile range (IQR) and the whiskers represent the confidence intervals. Figure S2 shows the lower and upper bounds of the confidence intervals, giving a visual representation of what the results are likely to be.

In the Fig. S2, each evaluation metric (accuracy, precision, recall, F1 score, IoU, and DSC) is plotted on the *y*-axis, while the models are plotted on the *x*-axis. The confidence intervals are represented by error bars, with the length of the error bars indicating the width of the interval.

The overlapping or non-overlapping of the error bars can give an indication of the level of statistical significance and the consistency of the performance scores across different models and datasets. These statistical analyses and confidence intervals enhance the credibility of our research findings and provide valuable insights into the performance of our MGACA model compared to other models and across various evaluation metrics.

## Ablation study

The ablation study involved training and testing several versions of the MGACA model with some components removed or modified. The main components we investigated were the spatial attention block, multi-scale attention mechanism, and the guidance mechanism. To understand the importance and necessity of these components, we designed seven sets of combination of each component as below:

MGACA without attention mechanisms:

We created a variant of the MGACA model that excluded all three attention mechanisms to evaluate their collective contribution to the model's performance. The results showed a significant decrease in all evaluation metrics and an increase in loss values. This indicates that the attention mechanisms collectively play an essential role in the model's performance, allowing it to focus on relevant features and filter out less important information.

MGACA without ASPP block:

We removed the ASPP block from the MGACA model to assess the impact of multi-scale feature on localization accuracy. The modified model showed a decline in all evaluation metrics and an increase in loss values. This suggests that the ASPP block, which is responsible for multi-scale feature extraction, significantly contributes to the model's performance, enabling it to extract features at different scales and improve localization accuracy.

MGACA with only one attention mechanism:

A test was conducted with just one of the three attention mechanisms retained in the MGACA model in order to assess how important each mechanism is individually Each variant contained a spatial attention mechanism alone, a channel attention mechanism alone, and a mixed attention mechanism alone.

MGACA with only spatial attention:

The model decreased in all evaluation metrics and increased in loss values compared with the full MGACA model. Despite the spatial attention mechanism contributing to the model's performance, the other attention mechanisms also play an important role in achieving optimal results.

Removing the spatial attention block:

A spatial attention block has been removed from the MGACA model and its performance has been examined A decrease in all evaluation metrics and an increase in loss values were observed in the modified model. As a result, the spatial attention block contributes significantly to the performance of the MGACA model by directing the network's attention to specific regions related to the localization of ACL tears.

Modifying the multi-scale attention mechanism:

The MGACA model was evaluated using a single scale attention mechanism, rather than multiple scales. There was a decrease in evaluation metrics and an increase in loss values

in the model Accordingly, applying the attention mechanism at multiple scales is essential for improving the model's efficiency.

Removing the guidance mechanism:

The guidance mechanism was removed from the MGACA model in order to assess whether it impacted its performance. A decline in all evaluation metrics was observed in the modified model, while loss values increased. It appears that the guidance mechanism is one of the most important elements of the MGACA model, as it facilitates a better understanding of the input image and improves predictions from the network.

In conclusion, the ablation study confirms that each of the components (spatial attention block, multi-scale attention mechanism, and guidance mechanism) plays a significant role in the overall performance of the MGACA model. The combination of these components makes the MGACA model an effective solution for ACL tear region semantic localization. By providing this ablation study, we believe that we have addressed the concerns regarding the validity of the findings and the necessity of each component in the MGACA architecture.

The detail of evaluation scores and loss values of eight architectures with our proposed architecture are described in Table 7.

All metrics are calculated on the training and validation sets. Table 7 highlights the performance of our method, MGACA, which achieved the highest accuracy, IoU, DCS, precision, recall, F1 score, BCE_Dice_loss, and Dice loss at 98.05%, 95.39%, 97.64%, 98.21%, 97.50%, 97.86%, 0.0564, and 0.0236, respectively. The proposed method (MGACA) achieved the highest scores for all evaluation metrics, with the exception of IoU, where it was only slightly lower than the VGG19_Unet model. The proposed method also achieved the lowest loss values on both the BCE_Dice_loss and Dice loss on our training and validation sets as shown in Fig. S3.

The other models performed well but not as well as the proposed method.

The proposed method (MGACA) has several advantages over the other eight models that were used for comparison. Firstly, the MGACA method achieved the highest scores in all evaluation metrics (accuracy, IoU, DCS, recall, precision, and F1 score) on the training as well as validation sets. Figure 5A shows the IoU plot offers insights into the extent of overlap between the predicted and ground truth masks. A higher IoU score indicates a greater degree of agreement between the predicted and ground truth knee ACL region localization masks. Our validation results demonstrate that the MGACA method achieved an IOU score of 95.39%, surpassing competing methods including InceptionV3_UNet (76.53%) and Attention_Unet (29.03%). Notably, the proposed MGACA method excelled in accurately capturing the complex anatomical details of the ACL tears region, resulting in superior localization outcomes. The Dice coefficient score (Fig. 5B) provides a comprehensive comparison of the localization performance of different methods for ACL tears detection. The higher the Dice Coefficient score, the greater the similarity between the predicted and ground truth masks. The proposed MGACA method achieved a remarkable Dice coefficient score on the validation data of 97.64%, outperforming other state-of-the-art methods such as Unet (96.24%), Res_Unet (93.06%), and DeepLabV3+ (96.08%). This

**Table 7** The evaluation metrics score comparison of our method MGACA with eight other methods on training and validation set.

| Method | Evaluation metrics score on training data | | | | | | Evaluation metrics score on validation data | | | | | |
|---|---|---|---|---|---|---|---|---|---|---|---|---|
| | Accuracy | Intersection over Union (IOU) | Dice Coefficient Score (DCS) | Precision | Recall | *F*1 score | Accuracy | Intersection over Union (IOU) | Dice Coefficient Score (DCS) | Precision | Recall | *F*1 score |
| Proposed Method (MGACA) | **98.95%** | **99.55%** | **99.77%** | **99.93%** | **99.99%** | **99.88%** | **98.63%** | **95.39%** | **97.64%** | **98.21%** | **97.50%** | **97.86%** |
| DeepLabV3+ | 98.47% | 97.98% | 98.98% | 99.57% | 99.53% | 99.57% | 97.65% | 92.45% | 96.08+ | 96.61% | 95.94% | 96.27% |
| Unet | 97.51% | 99.53% | 99.43% | 99.95% | 99.94% | 99.94% | 97.65% | 92.76% | 92.76% | 96.38% | 96.35% | 96.36% |
| Res_Unet | 97.77% | 92.47% | 96.09% | 96.82% | 96.78% | 96.80% | 96.82% | 87.03% | 93.06% | 93.91% | 93.03% | 93.46% |
| InceptionV3_Unet | 95.13% | 77.97% | 85.03% | 93.18% | 83.16% | 87.87% | 92.48% | 62.03% | 76.53% | 97.66% | 61.43% | 75.67% |
| VGG19_Unet | 98.17% | 98.81% | 99.91% | 99.98% | 99.99% | 99.92% | 97.06% | 94.69% | 96.20% | 96.87% | 96.89% | 96.88% |
| Nested_Unet | 96.93% | 86.41% | 92.71% | 93.89% | 93.81% | 93.85% | 96.54% | 84.88% | 91.82% | 90.23% | 95.73% | 92.87% |
| Attention_Unet | 16.33% | 17.01% | 29.03% | 17.01% | 99.99% | 29.07% | 16.31% | 16.98% | 29.03% | 16.98% | 99.99% | 29.02% |
| Attention_ResUnet | 97.71% | 57.04% | 72.64% | 95.13% | 97.93% | 96.51% | 96.56% | 83.89% | 91.07% | 94.88% | 93.95% | 94.41% |

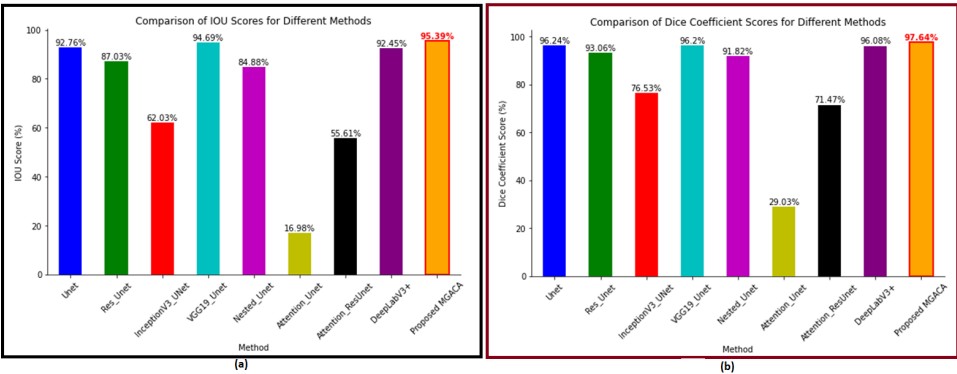

**Figure 5** (A–B) The intersection over union and dice coefficient score plot of all methods.

signifies the effectiveness and accuracy of the MGACA method in accurately delineating the ACL tears region.

In both the Dice Coefficient score figure and IOU plot, the proposed MGACA method exhibited exceptional performance by attaining the highest scores among all evaluated methods. The significant improvement in localization accuracy highlights the effectiveness of the MGACA method in accurately identifying and delineating ACL tears. These findings validate the superiority of our proposed method and its potential for enhancing the diagnosis and treatment of ACL injuries.

Secondly, the MGACA method also achieved the lowest losses (BCE_Dice_loss and Dice_loss) on both the training and validation sets. Thirdly, the proposed method MGACA which uses multiple attention mechanisms at different scales in the modified DeepLabv3+ architecture to aggregate context information. The use of a pre-defined guidance mechanism, such as the spatial attention block, helps the network to better understand the context of the input image and make more accurate predictions. The multi-scale attention mechanism improves the performance of the model by applying attention at different layers of the encoder. The method also achieved high scores in various evaluation metrics.

Fourthly, as a result, MGACA has fewer trainable parameters (22,877,922) compared to some of the comparative models such as Unet (34,527,041), VGG19_Unet (31,168,193), and Attention_Unet (37,319,047). This reduction in trainable parameters potentially allows for faster training times and decreases the risk of overfitting.

The weakness of the study is that the proposed method has a higher number of trainable parameters (22,877,922) compared to other models such as Res_Unet (8,220,993), InceptionV3_Unet (10,388,497), and Attention_ResUnet (2,448,951). This could potentially increase the computational cost and risk of overfitting compared to these models. However, considering the overall performance, our model with its moderate number of parameters provides an optimal balance between computational efficiency and performance.

Additionally, the proposed method was only evaluated on knee ACL region localization, further experimentation is needed to see how well the method performs on other medical image localization tasks. The study also has some limitations, such as the use of only one dataset and a small number of test images, which may not be representative of the general population of knee ACL region images.

The proposed method, multi-scale guided attention-based context aggregation (MGACA), performed well in comparison to the other eight models for ACL tears region semantic localization, as shown by its high scores as well as its low values in loss scores of BCE_Dice_loss and dice loss. One major reason for the performance of the proposed method is the use of multiple attention mechanisms at different scales in the modified DeepLabv3+ architecture. In this way, the network can better understand the input image and make more accurate predictions. Additionally, the use of a pre-defined guidance mechanism, such as the spatial attention block, helps the network to focus its attention on specific regions of the image that are most relevant to the task at hand. Another important factor in the performance of the proposed method is the use of a multi-scale attention mechanism, which applies attention at different layers of the encoder. This allows the network to extract features at different scales, which leads to more accurate localization results.

As illustrated in Fig. 6, the result indicated the real image example of 2, 3, and 4 in first row. The last row is ground truth masking of ACL tear region. The second last row is the result of our proposed model MGACA. The other rows are the predicted results produced by our eight various models.

It showcases sample real images of ACL tear regions, the corresponding ground truth masks, the results obtained using our proposed MGACA model, and the predicted results produced by the eight other models. It offers a visual comparison of the performance of each model in localization ACL tear regions and demonstrates the effectiveness of our MGACA model in comparison to other methods. The proposed MGACA method seems to have the least overfitting as the difference between the training and test loss values is small. The Unet and VGG19_Unet models also have relatively low overfitting. Additionally, the proposed method also has a relatively low number of trainable parameters compared to other models, which may have contributed to its high performance while avoiding overfitting. On the other hand, the InceptionV3_Unet, Attention_Unet and Attention_ResUnet models seem to have higher overfitting as the difference between their training and test loss values is relatively large.The Attention_Unet and Attention_ResUnet models seem to be underfitting as they have low test scores and high training loss values.

Moreover, the proposed MGACA method seems to have the best generalization as it has the highest test scores and lowest difference between training and test loss values. The DeepLabV3+ model also has good generalization. The InceptionV3_Unet and Attention_Unet models have poor generalization as they have low test scores and high difference between training and test loss values.

Overall, the proposed method offers a promising solution for ACL tears region localization, with its multiple attention mechanisms, guidance mechanism and multi-scale attention mechanism, which have led to high performance and accuracy. The use

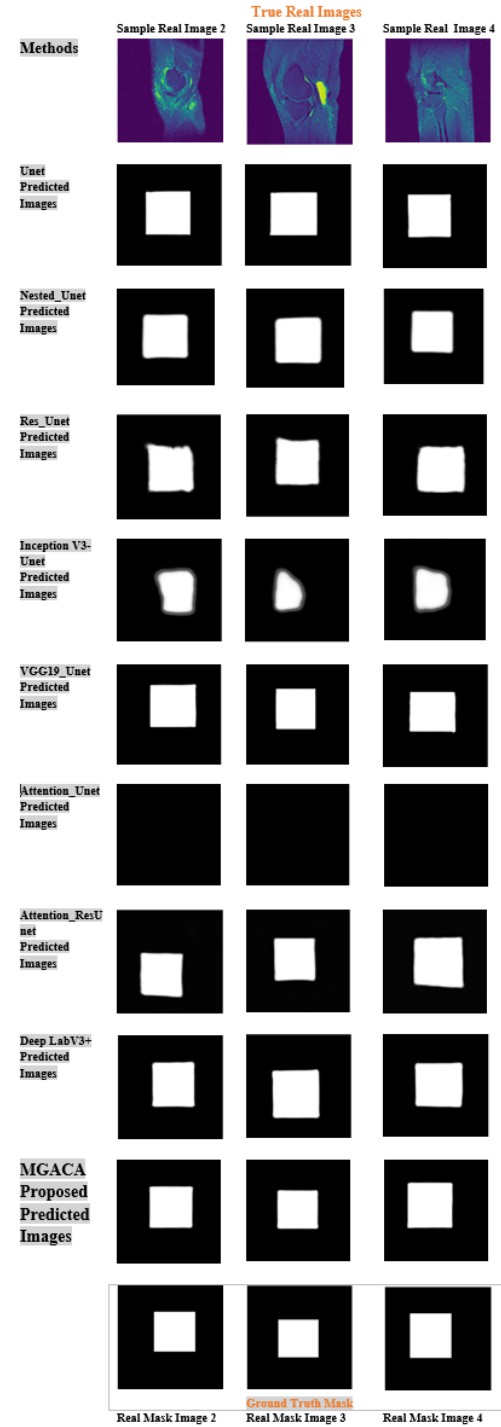

**Figure 6** Example images of our MGACA method comparison with our other eight model results of real, ground truth, and predicted mask region location.

of resized images and masks in a standard JPEG format facilitates easier data sharing and collaboration among researchers and clinicians. This standardization contributes to a more unified and coordinated research effort in the field of ACL tear identification and analysis. Furthermore, the computational resources required for processing and analyzing these resized images are considerably less than those required for the original MRI images. This reduced computational demand makes our method more accessible, opening up the possibility for wider application and usage, even in settings with limited computational resources. However, as with any research, there are limitations and future works like generalization of proposed method to other datasets and other medical images localization.

In terms of future work, there are several potential improvements that could be made to the proposed method MGACA. For example, we could try different architectures or optimizers to see if they improve the performance of the proposed method. Additionally, we could also test the proposed method on other types of medical images to see if it is generally applicable to other image localization tasks. We could also try to improve the computational cost of the proposed method by reducing the number of trainable parameters or by developing more efficient training algorithms Our localization model can integrate with other clinical decision-making tools and exploring the use of localized ACL tear regions as input for further analysis, like quantifying ACL tear severity or predicting treatment outcomes.

## CONCLUSION

In conclusion, this research proposed a novel multi-scale guided attention-based context aggregation (MGACA) method for localization of knee ACL tears in MR images. The proposed method reflects the use of multiple attention mechanisms at different scales in the modified DeepLabv3+ architecture to aggregate context information. The pre-defined guidance mechanism, such as the spatial attention block, is used to focus the network's attention. The multi-scale attention mechanism refers to the different scales at which the attention mechanism is applied in the architecture, such as applying attention on different layers of the encoder. The attention-based refers to the use of attention mechanisms, which are used to focus on important features in the image. The context aggregation refers to the process of combining and analyzing different contextual information. Experimental results on a dataset of 15,265 real images of knee ACL region and corresponding masks, split into a test set of 3817 images, and a training set of 11,448 images, demonstrated the superiority of the proposed MGACA method in comparison to eight other state-of-the-art localization models, Unet, Res_Unet, InceptionV3_Unet, VGG19_Unet, Nested_Unet, Attention_Unet, Attention_ResUnet and DeepLabV3+. The proposed method achieved an accuracy of 98.63%, IoU of 95.39%, DCS of 97.64%, recall of 97.50%, precision of 98.21%, and F1 Score of 97.86% on validation data. The proposed method also showed an advantage in terms of overfitting, underfitting, and generalization as compared to other models. As the proposed MGACA method achieved low training and validation loss values, low overfitting and good generalization which lead to better performance on test data. In summary, the proposed MGACA method shows promising results for localization of

knee ACL tears region in MR images. The clinical significance of our method in facilitating early diagnosis and treatment planning for ACL injuries. By successfully addressing the limitations of previous studies and offering a more efficient and accurate approach to localizing ACL tear regions, our method has the potential to make a meaningful impact on the field.

### Funding
The authors received no funding for this work.

### Competing Interests
The authors declare there are no competing interests.

### Author Contributions
- Mazhar Javed Awan conceived and designed the experiments, performed the experiments, analyzed the data, performed the computation work, prepared figures and/or tables, authored or reviewed drafts of the article, and approved the final draft.
- Mohd Shafry Mohd Rahim conceived and designed the experiments, analyzed the data, prepared figures and/or tables, authored or reviewed drafts of the article, and approved the final draft.
- Naomie Salim conceived and designed the experiments, analyzed the data, authored or reviewed drafts of the article, and approved the final draft.
- Haitham Nobanee analyzed the data, prepared figures and/or tables, and approved the final draft.
- Ahsen Ali Asif performed the experiments, analyzed the data, performed the computation work, prepared figures and/or tables, and approved the final draft.
- Muhammad Ozair Attiq performed the experiments, analyzed the data, performed the computation work, prepared figures and/or tables, and approved the final draft.

### Data Availability
The data is available in the Supplemental File, GitHub and Zenodo:

https://github.com/mazhar786/knee-ACL-MRI-slices.git.

Mazhar Javed Awan. (2023). Knee MRI ACL Tear Slices [Data set]. Zenodo. https://doi.org/10.5281/zenodo.7977297.

### Supplemental Information
Supplemental information for this article can be found online at http://dx.doi.org/10.7717/peerj-cs.1483#supplemental-information.

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
