# Peer review of "MGACA-Net: a novel deep learning based multi-scale guided attention and context aggregation for localization of knee anterior cruciate ligament tears region in MRI images"

_PeerJ Computer Science, doi:10.7717/peerj-cs.1483_

## Round 0.1 · original submission · Major Revisions

The reviewers have substantial concerns about this manuscript, especially about the significance of this specific topic as well as the details of the proposed algorithm. The authors should provide point-to-point responses to address all the concerns and provide a revised manuscript with the revised parts being marked in different color.

Reviewer 1 ·

Basic reporting

The manuscript is well-structured and clearly presents the problem of manual segmentation of the ACL in MRI images and the proposed solution using deep learning techniques. The inclusion of multiple evaluation metrics adds to the credibility of the study and provides a comprehensive understanding of the model's performance. Moreover, the findings suggest that MGACA provides an accurate and efficient solution for automating the segmentation of the ACL in knee MRI images, surpassing other state-of-the-art models in terms of accuracy and loss values.

The manuscript presents a well-designed study and provides valuable contributions to the field of ACL tear diagnosis using deep learning techniques. However, I also identified several weaknesses.

It would be helpful to move the experimental setup and evaluation metrics from the Results section to the Material & Methods section. This would make the manuscript easier to follow and allow the reader to better understand the research methodology.

The Results section should include some description of the results, such as tables or figures, to better illustrate the findings of the study. The manuscript contains several tables and figures, but they are not well annotated. It would be beneficial if the authors included a brief annotation of each table and figure to help the reader understand.

Experimental design

In the background section, the authors mentioned several limitations of previous studies, including limited sample sizes, limited training datasets, manual segmentation, and long processing time. It is not entirely clear how the MGACA method addresses these limitations, and this should be better explained

Validity of the findings

The MGACA architecture contains several unique components, including three types of attention mechanisms and an ASPP block. It would be helpful if the authors could perform ablation studies to demonstrate the importance and necessity of each component.

Reviewer 2 ·

Basic reporting

The manuscript proposed a deep learning network to segment knee Anterior cruciate ligament (ACL) tears region in MRI images. While I applaud the author’s efforts, I do not believe that the paper is suitable to publish on PeerJ. The main reasons for my decision are as follows.

(1) It is less significant to use such a relatively rectangular mask as the ground truth to guide the ACL segmentation model. Although the claimed accuracy is quite high, no clinical meaning to the doctors to decide the treatment plan.
(2) The segmentation model is not for the MRI image. The authors turned the MRI image to grayscale JPEG image. This make the segmented result meaningless. Again, the segmentation on the JPEG image cannot offer clinical guide. Cannot calculate the area of the ACL tears region on the JPEG image
(3) It is not suitable to claim the dataset as 15268 knee MRI images. In fact, it is 917 MRI images, resulting 15268 slices
(4) The authors need to clearly explain how they trained the model, eg. How to split the train set and the valid set, what is the mini-batch size, what optimum algorithm is used in the model training, etc.
(5) Most of equations are non-standard
(6) Page 13 Line 350 should be “The equation of IOU is:”
(7) Table 4 and Table 6 are figures, not tables
(8) It is highly unreliable that there are no changes in Dice coefficients and IOU as epochs increase.

Experimental design

(1) It is less significant to use such a relatively rectangular mask as the ground truth to guide the ACL segmentation model. Although the claimed accuracy is quite high, no clinical meaning to the doctors to decide the treatment plan.
(2) The segmentation model is not for the MRI image. The authors turned the MRI image to grayscale JPEG image. This make the segmented result meaningless. Again, the segmentation on the JPEG image cannot offer clinical guide. Cannot calculate the area of the ACL tears region on the JPEG image
(3) The authors need to clearly explain how they trained the model, eg. How to split the train set and the valid set, what is the mini-batch size, what optimum algorithm is used in the model training, etc.

Validity of the findings

It is highly unreliable that there are no changes in Dice coefficients and IOU as epochs increase.

Reviewer 3 ·

Basic reporting

The authors present a novel deep learning algorithm for accurately segmenting the ACL in MRI images. They provide a clear explanation of the problem and limitations associated with the current methods used for ACL segmentation and propose a solution to address this issue. A comprehensive review of the existing research in this area is also provided. The proposed algorithm incorporates multi-scale information and guided attention mechanisms to accurately capture the complex knee anatomy and produce precise segmentation predictions. The authors evaluate the algorithm using a sufficient dataset and demonstrate that it outperforms other state-of-the-art algorithms, indicating its potential to improve clinical diagnosis and treatment of ACL injuries. The manuscript is well written in professional English throughout. The figures and tables are clear to read and understand. The algorithms proposed have scientific novelty.

However, one major concern is the annotation of the ground truth mask. It is noted that the definition of the ground truth mask is ambiguous and arbitrary, which might induce large bias in the image segmentation and that the evaluation metrics used in comparing results may be affected by ground truth delimitation. Additionally, while the authors describe the architecture of the network and explain the purpose of each key components, it is not entirely clear why these specific methods were chosen. Although the authors emphasize that the ASPP block and multi-attention can better capture the contextual information, and other key components have other advantages, the experiments and data do not provide sufficient evidence to support this claim.

Experimental design

A major concern regarding the experiment design is the methodology used to generate the ground truth mask segmentation. Typically, in clinical MRI reading, a highly experienced radiologist is required to conduct the segmentation and annotation according to clinical criteria, with repetition for better accuracy. Furthermore, recent studies and literature reviews cited by the authors in the background section also use more precise segmentation as ground truth. However, in this study, the ground truth mask shown in Figure 5 covers a very large square region and is not specific to the knee anatomy structure. Because of that, the context of the image was not reflected in any step of the segmentation process, raising questions about the necessity of the complicated architecture of the deep neural network. Any changes like size or positional movement to this mask may significantly affect the segmentation results.

For image analysis and preprocessing in MRI sequence volumes data, there are a few questions and adding more details to the manuscript will be helpful.
1. MRI sequences usually contain many slices, how do you choose slices with labels (meaning image slices with sufficient keen regions captured).
2. How do you resize the images and decide the dimension? Is there any difference in the patients characteristic like age, gender that might induce variance to the images? Are there any steps of registration for standardization? Is there image augmentation to enhance the dataset?
3. Are there any control images without ACL injures and if so, is there any difference in processing the samples?

For deep neural network and implementation:
1. How is the train/ validation/test split? Table 4 shows the performance for training and validation set but the discussion section talks about test set.
2. For evaluation and comparison with other methods, how are those methods trained? Was there any parameter tuning steps to make sure these methods are optimized to the best performance on this dataset before comparing with the proposed new algorithm?
3. In Figure 4 and 5, the dice coefficient score and the intersection over union score don’t change over epoch, which might indicate the model is not optimizing – What is the optimizing method and the learning rate? Is there any approach you adapted to avoid overfitting?
4. Line 402 says the strength of this method is that it has less number of trainable parameters as compared to other models. Line 415 has the opposite conclusion that this weakness of the study is the higher number of trainable parameters.

Some additional suggests:
The “Results” section with Experiment Setup and Evaluation Metrics should be move to the ‘Materials & Methods” section.

The section of algorithm and segmentation methodology contains some repeated and generalized description of each algorithm, which can be more precise with more details. E.g. Line 228 – 235 how does the context filtered? Line 260 -261 repeats three attention mechanisms. Line 251, it’s better to use consistent “ReLU” instead of “relu”.

Validity of the findings

1. Although the new neural network architecture is very sophisticated designed, there is no data show how much performance gain can be achieved by adopting each part of these component besides the dice coefficient and intersection over union scores.
2. The label of ground truth might induce bias for the performance comparison.
3. It would be better to include some statistics analyses of the results.

---

## Round 0.2 · accepted · Accept

Concerns have been addressed and I recommend accepting the manuscript.

Reviewer 1 ·

Basic reporting

The author revised manuscript fulfilling the comments.

Experimental design

The author revised manuscript fulfilling the comments.

Validity of the findings

The author revised manuscript fulfilling the comments.